# Thyroid Dysfunction in Advanced Heart Failure Patients and Its Correlation with Amiodarone Therapy: A Two-Year Study

**DOI:** 10.3390/biomedicines12030567

**Published:** 2024-03-03

**Authors:** Agnieszka Kuczaj, Szymon Warwas, Anna Danel, Piotr Przybyłowski, Tomasz Hrapkowicz

**Affiliations:** 1Department of Cardiac, Vascular and Endovascular Surgery and Transplantology, Faculty of Medical Sciences in Zabrze, Medical University of Silesia, 40-055 Katowice, Poland; szymonwarwas@gmail.com (S.W.); thrapkowicz@sum.edu.pl (T.H.); 2Silesian Centre for Heart Disease, ul. M. C. Skłodowskiej 9, 41-800 Zabrze, Poland; 3Department of Lung Diseases and Tuberculosis, Faculty of Medical Sciences in Zabrze, Medical University of Silesia, 40-055 Katowice, Poland

**Keywords:** heart failure, thyroid dysfunction, amiodarone, TSH

## Abstract

(1) Background: Advanced heart failure often accompanies ventricular arrhythmias, necessitating antiarrhythmic therapies. Amiodarone, commonly used for this purpose, may induce thyroid dysfunction due to its high iodine content. However, the prevalence and correlation of thyroid dysfunction with amiodarone in end-stage heart failure patients remain unclear. Aim: This study aimed to evaluate the prevalence and types of thyroid dysfunction and their association with amiodarone among 200 patients diagnosed with advanced heart failure eligible for transplantation. (2) Methods: Consecutively enrolled patients received treatment following the European Society of Cardiology guidelines and were followed-up for two years. Ventricular arrhythmias affected 58.5% of the cohort, with 24.5% receiving amiodarone therapy. (3) Results: Thyroid metabolism dysfunction was evident in 61 patients, notably overrepresented in women (*p* = 0.0028). Hyperthyroidism (34 patients) and hypothyroidism (27 patients) were observed, with a significant amiodarone-related correlation. Despite this, thyroid dysfunction was not associated with increased mortality among the studied group. (4) Conclusions: Thyroid dysfunction is prevalent in advanced heart failure patients, with a notable proportion linked to amiodarone. However, its presence does not correspond to higher mortality rates. Understanding these associations is crucial for effective management in this patient population. Further exploration is warranted to refine approaches to thyroid dysfunction in refractory heart failure.

## 1. Introduction

Heart failure (HF) has been defined as a clinical syndrome with symptoms and/or signs caused by a structural and/or functional cardiac abnormality and corroborated by elevated natriuretic peptide levels and/or objective evidence of pulmonary or systemic congestion (universal definition) [1]. HF affected more than 64 million individuals worldwide in 2017. Due to its widespread impact, it has been classified as a global pandemic [2]. The anticipated rise in the incidence of HF can be attributed to the availability of evidence-based treatments that improve survival rates. This, coupled with a longer life expectancy among the general population, leads to an increased prevalence of the condition. Heart transplantation, along with left ventricular assist device implantation, is the sole treatment option for patients diagnosed with end-stage heart failure [3]. According to research conducted by the Ministry of Health, the average waiting time for a heart transplant in Poland between 2011 and 2020 was 603 days [4]. Despite significant advances in medical treatment, the prognosis for patients suffering from HF remains unfavorable. In a registry of the European Society of Cardiology, one-year mortality rates ranged from 21.6% to 36.5% for acute HF patients and from 6.9% to 15.6% for chronic HF patients across different European countries [5]. It has been observed that sudden and unexpected deaths in patients with HF constitute up to 50% of overall fatalities. The underlying cause of sudden cardiac death is generally attributed to a lethal arrhythmia, most frequently ventricular tachycardia or ventricular fibrillation [6]. Numerous randomized controlled trials have demonstrated the crucial role of implantable cardioverter-defibrillators (ICDs) in preventing sudden cardiac death (SCD) among HF patients with a left ventricular ejection fraction (LVEF) of 35% or less [7,8]. ICD recipients who experience frequent and symptomatic ventricular tachycardia (VT) if catheter ablation is not a therapeutic option should be treated medically with antiarrhythmic drugs, including amiodarone [9,10].

Amiodarone stands out as one of the most frequently used antiarrhythmic drugs [11,12]. Categorized as a class III antiarrhythmic agent according to the Vaughan Williams classification, it exerts an inhibitory effect on potassium channels, leading to the prolongation of the third phase of the action potential in the conducting cells of the heart [12]. Despite its broad spectrum of applications and positive treatment outcomes, amiodarone is associated with a range of adverse effects [13]. Significantly, attention is drawn to the frequently observed changes in thyroid function tests, but also to the impairment of thyroid function associated with the presence of iodine in the chemical structure of amiodarone [14].

The indispensable significance of the proper functioning and maintenance of thyroid hormones (THs) resonates across both biological and medical domains. THs wield a profound influence on the growth and development of nearly every cell and organ, assuming critical roles during embryogenesis, early life, and throughout adulthood [15]. THs also induce significant metabolic effects, including alterations in the metabolism of lipids, proteins, carbohydrates, vitamins, and oxygen consumption [16]. Exerting a substantial influence, THs play a pivotal role in the development and functioning of the cardiovascular system. This direct impact extends to the entirety of the circulatory system while indirectly affecting the systemic vasculature, the renin–angiotensin–aldosterone system, and the autonomous nervous system [17]. THs also exhibit non-genomic effects that entail the rapid transport of ions, including calcium, sodium, and potassium, across the plasma membrane, along with the facilitation of glucose and amino acid transport. Furthermore, THs modulate mitochondrial function and the orchestration of diverse intracellular signaling pathways [18,19]. Activated through the phosphatidylinositol 3-kinase (PI3K)/serine/threonine protein kinase (AKT) signaling pathways, thyroid hormones (THs) induce the production of endothelial nitric oxide, resulting in ensuing vasodilation [20]. In the adult heart, THs exhibit a pro-angiogenic effect, capable of inducing arteriolar growth in both normal cardiac conditions and following myocardial infarction [21,22].

Hypothyroidism is typified by the diminished levels of thyroid hormones T3 and T4 alongside elevated levels of TSH. The clinical condition can manifest with sinus bradycardia, atrioventricular blocks, and the prolongation of the QT interval, raising the risk of dangerous ventricular arrhythmias like torsades de pointes [23,24]. Hypothyroidism should be considered in the differential diagnosis of polymorphic ventricular tachycardia (VT) [25].

Severe hypothyroidism may coexist with ischemic heart disease, decreased myocardial contractility, pericardial effusion, decreased cardiac output, and high blood pressure [26]. The correction of thyroid hormone levels is necessary for ventricular tachycardia and ventricular fibrillation associated with hypothyroidism.

Hyperthyroidism is linked with increased automaticity in the pacemaker system and triggered activity in the atria and ventricles, potentially leading to tachyarrhythmias [27]. A common rhythm disturbance in hyperthyroidism is supraventricular ectopic beats, which typically resolve with thyroid dysfunction treatment [27]. Even subclinical hyperthyroidism (TSH levels below the lower limit of normal, and fT4 levels within the normal range) is associated with a higher resting heart rate and an increased incidence of supraventricular and ventricular premature contractions [28].

Although hyperthyroidism commonly leads to tachyarrhythmias, it can also, in rare cases, result in bradyarrhythmias [29]. A rare complication like complete atrioventricular block associated with hyperthyroidism typically occurs in patients with additional risk factors such as hypercalcemia, infectious diseases, rheumatic fever, or digoxin intake [29].

The study conducted by Mitchell et al. involved a cohort of 2225 individuals and aimed to establish a correlation between thyroid abnormalities in patients with heart failure and their risk of death [30]. The findings of the study revealed that deviations in thyroid function among patients with symptomatic heart failure and ejection fraction (EF) below 35% were associated with increased mortality risks. Both hypo- and hyperthyroid states showed higher relative risks of mortality compared to a normal thyroid state. The respective increments were 58% and 85%, with *p* < 0.0001 for hypothyroid and *p* = 0.0048 for hyperthyroid compared to a normal thyroid state [30].

To ensure precision, we adopt the definitions of hyperthyroidism and thyrotoxicosis provided by De Leo et al. Hyperthyroidism is characterized by the elevated production and release of thyroid hormones by the thyroid gland, while thyrotoxicosis refers to the clinical condition marked by an excess of circulating thyroid hormones, regardless of their origin [31].

Treatment with amiodarone may lead to the development of amiodarone-induced hypothyroidism or hyperthyroidism, as well as amiodarone-induced thyrotoxicosis (AIT) [14].

The risk of thyroid function disorders increases approximately fivefold among patients treated with amiodarone [32,33]. The overall frequency of AIT based on the literature data ranges from 1% to 23% of patients and is three times more prevalent in males than females [14,33,34]. Observations indicate that AIT can manifest suddenly, either in the early stages of amiodarone therapy or after prolonged use [34]. The likelihood of AIT increases with the duration of amiodarone treatment. The average duration of amiodarone therapy before detecting AIT is approximately 3 years, identifying the subgroup of advanced heart failure patients receiving chronic amiodarone as the most endangered group. Due to the storage of the drug and its metabolites in tissues and the slow release of amiodarone, the risk of AIT development persists even several months after treatment discontinuation [35].

An increased incidence of AIT has been noted in geographical areas characterized by insufficient iodine intake [35]. AIT can be classified into two types: AIT type 1 and AIT type 2 [14]. AIT type 1 arises from excessive thyroid hormone synthesis in patients with latent thyroid disease (e.g., Graves’ disease, nodular goiter). AIT type 2 results from a developing, destructive, inflammatory process involving a previously healthy thyroid gland. In this type, thyroid follicles are destroyed, leading to the release of thyroid hormones into the bloodstream [36,37,38].

Amiodarone-induced hypothyroidism (AIH) typically occurs during amiodarone therapy in 1–32% of patients [39]. AIH is more prevalent in women and elderly patients compared to men [40]. There is a correlation between the occurrence of AIH in patients with previously undiagnosed autoimmune thyroiditis, such as Hashimoto’s disease [41]. AIH is more common in regions with sufficient iodine consumption [40,41]. Clinical symptoms of AIH do not differ from those of hypothyroidism with other etiologies [40]. Additionally, the presence of anti-thyroid peroxidase antibodies is a significant factor in the pathogenesis of AIH, and the combination of female gender with the simultaneous presence of anti-thyroid antibodies leads to a 13.5-fold higher likelihood of AIH [42]. Severe hypothyroidism may predispose to the occurrence of torsades de pointes. The diagnosis of overt AIH is based on blood biochemical tests indicating low FT4 levels and high TSH levels in serum [40]. The treatment regimen involves the use of L-thyroxine in patients diagnosed with AIH, and the complete discontinuation of amiodarone is not deemed necessary [43].

The precise prevalence of thyroid dysfunction among patients with end-stage heart failure treated with amiodarone remains unclear. We conducted a study to investigate the prevalence and types of thyroid dysfunction among patients diagnosed with advanced heart failure.

## 2. Materials and Methods

We enrolled 200 patients diagnosed with advanced heart failure according to the current ESC criteria [44]. The patients were admitted consecutively between 31 December 2018 and 31 December 2019, to the Department of Cardiac Transplantation and Mechanical Circulatory Support in order to qualify them for heart transplantation. In the event that the fulfillment of advanced heart failure, as per the ESC criteria, was confirmed during the hospital evaluation, we enrolled the patients for further analysis. They underwent regular follow-ups every six months, following the heart failure surveillance protocol implemented in the ward. We assessed the patients over a two-year period since their enrollment. Survival was assessed using data from the national insurance system in addition to hospital records.

We analyzed the cause of heart failure, biometric values, comorbidities, applied medication, the presence of ventricular arrhythmia, two-year mortality, and transplantation rate retrospectively. The data were acquired at inclusion from patients’ clinical documentation during hospital stays, ambulatory visits, and cardioverter-defibrillator controls. The analysis was aimed to assess the presence of thyroid dysfunction in the group of advanced heart failure patients. The data concerning thyroid dysfunction were analyzed from individual patient documentation and diagnoses. Furthermore, based on current laboratory diagnostics, we defined euthyreosis as a thyroid-stimulating hormone (TSH) level between 0.35 and 4.70 mIU/L, hyperthyroidism as TSH < 0.35 mIU/L, and hypothyroidism as TSH > 4.70 mIU/L. In the case of abnormal TSH values, the additional analysis of FT3 and FT4 was conducted. We investigated the type of thyroid dysfunction and its association with amiodarone administration and ventricular arrhythmia. Ventricular arrhythmias were assessed based on Holter ECG findings and implantable cardioverter-defibrillator controls. The cardioverter-defibrillator controls were performed every six months. Additional ECG Holters were performed in the case of any doubts or uncertainties. Non-sustained ventricular tachycardia (VT) was defined as VT lasting shorter than 30 s, while sustained VT was defined as VT lasting longer than 30 s or aborted by ICD.

### Statistical Analysis

The presentation of categorical variables utilized counts and percentages, while continuous variables were expressed as the mean and standard deviation or median with lower and upper quartiles. The normal distribution of data was assessed using the Shapiro–Wilk test. Categorical variables were compared using the chi-square test, and continuous variables were compared using the *t*-test or Mann–Whitney U test as appropriate.

A significance level of *p* < 0.05 was considered statistically significant. SAS software version 9.4 (SAS Institute Inc., Gary, NC, USA) was employed for all calculations.

## 3. Results

The mean age of the analyzed patients was 56.5 (±10) years, with women comprising only 16% of the study group (32 patients). As of 31 December 2021, the all-cause mortality rate was 36.5%, and within the entire analyzed group, 48 patients underwent heart transplantation (24%). Patients who underwent heart transplantation were analyzed in the study up to the time of transplantation. Treatment protocols for all patients adhered to the current guidelines set by the European Society of Cardiology for the treatment of heart failure [44,45]. Baseline clinical characteristics are presented in Table 1.

In the studied group, 75% (150) received angiotensin-converting enzyme inhibitors (ACEIs)/angiotensin receptor blockers (ARBs), 12.5% (25) were administered an angiotensin receptor-neprilysin inhibitor, 94% (188) received a mineralocorticoid receptor antagonist, and 97.5% (195) were prescribed beta-blockers. Throughout the evaluation period, 24.5% (49) of patients underwent amiodarone therapy, and a history of amiodarone usage (current or past) was identified in 65 patients.

Thyroid hormone supplementation was required in 25 (12.5%) patients, and 15 (7.5%) required the application of thiamazole. Overall, 17 (8.5%) patients had TSH level outside of the 0.35–4.94 µIU/mL range, with 5 below and 12 above this recommended range. In the case of FT3, 7 patients exhibited results beyond the range of 1.580–3.910 pg/mL, and among them, 7/7 (100%) exhibited levels above the normal range. FT4 results were outside of the recommended range, i.e., 0.7–1.48 ng/dl, in 12 patients, of which 11 (91.67%) had results above and 1 (8.33%) below the reference values. The mean value of TSH [SD] was 2.2 [2.74], of FT3 was 2.47 [0.47], and of FT4 was 1.12 [0.19].

In total, 61 (30.5%) patients exhibited thyroid dysfunction. Among the deceased patients, 23 out of 73 (31.5%) had thyroid disorders. In the subgroup of deceased patients, 12 were diagnosed with hyperthyroidism and 11 with hypothyroidism; however, no significant differences were found in terms of mortality (*p* = 0.81). Comprehensive details regarding mortality related to thyroid dysfunction are presented in Table 2.

We observed a significant discrepancy between genders. Thyroid dysfunction was present among 46.9% (15/32) women and 27.4% (46/168) men; *p* = 0.028. In the population of women, hypothyroidism predominated, with 13 out of 15 patients diagnosed with hypothyroidism, compared to only 2 out of 15 patients diagnosed with hyperthyroidism. In the male subpopulation, hyperthyroidism was more prevalent, with 32 out of 46 patients diagnosed with hyperthyroidism, compared to 14 out of 46 patients diagnosed with hypothyroidism; *p* = 0.00014. In the female population, there were no cases of hyperthyroidism or hypothyroidism attributed to amiodarone. Among males, amiodarone resulted in 20 out of 32 cases of hyperthyroidism and 1 out of 14 cases of hypothyroidism.

In total, 34 patients were diagnosed with hyperthyroidism, with 14 having hyperthyroidism not directly linked to the application of amiodarone and 20 as a consequence of administering amiodarone, and 30.1% of the entire population of heart failure patients who received amiodarone (20 out of 65 patients). In contrast, 27 patients were diagnosed with hypothyroidism, encompassing 26 patients in whom it was not caused by the use of amiodarone and 1 case where it was attributed to amiodarone; *p* < 0.001. Overall, in 34.4% of patients, thyroid dysfunction was a consequence of administered antiarrhythmic treatment, constituting 10.5% of the study population with heart failure.

## 4. Discussion

Over the last 20 years, significant efforts have been dedicated to examining the impact of thyroid hormones on the cardiovascular system. This scrutiny has specifically extended to discerning the potential effects of these hormones on diverse outcomes in individuals facing HF [46]. Pingitore et al. explored that T3 serum concentrations might independently predict both left ventricular (LV) dysfunction and the functional class assigned by the New York Heart Association [47]. According to Gerdes et al. and Cappola et al., thyroid dysfunction is emerging as a modifiable risk factor for individuals who are at risk of developing heart failure. This is supported by the observation that even minor changes in the concentration of circulating thyroid hormones can have deleterious effects on the cardiovascular system [48,49,50].

As a result of the substantial physiological role that THs play in cardiovascular homeostasis, affecting cardiac output, contractility, vascular resistance, and blood pressure, clinical outcomes for patients with HF are notably affected by any disruptions in TH levels or function [51,52]. Overt and subclinical hypo- and hyperthyroidism both impact cardiac function through intricate mechanisms, potentially resulting in cardiac dysfunction and the onset of heart failure (HF) [53]. Furthermore, both hypothyroidism and hyperthyroidism have the potential to contribute to heart failure, with a more pronounced impact on previously damaged myocardium [50]. THs induce a reduction in systemic vascular resistance via vasodilation, and by positive chronotropic and inotropic effects, they elevate cardiac output. In the realm of blood pressure, both hyperthyroidism and hypothyroidism are associated with hypertension. Specifically, hyperthyroidism tends to preferentially elevate systolic arterial pressure, while hypothyroidism leads to an increase in diastolic and median arterial pressure [54]. Over time, hyperthyroidism may induce myocardial hypertrophy reminiscent of the hypertrophic changes observed during pregnancy. This is characterized by an augmentation in both the length and cross-section of cardiomyocytes, as evidenced by studies by Fazio et al. and Biondi et al. [18,55]. Subclinical and overt hypothyroidism are common occurrences in individuals with congestive heart failure (CHF), potentially exacerbating the compromise of cardiac pump efficiency [52,56]. Subclinical hypothyroidism has been linked to left ventricular diastolic dysfunction at rest and during exertion, as well as impaired left ventricular systolic function during exercise. Higher TSH levels in patients with subclinical hypothyroidism have been associated with a decrease in left ventricular stroke volume and cardiac index and an increase in systemic vascular resistance [57].

Patients with TSH levels falling within the range indicative of primary hypothyroidism exhibited a heightened risk of acute events associated with CHF, including hospital admissions due to acute decompensation and death [58]. Iacoviello et al. identified a significant association between overt hypothyroidism and an increased risk of mortality and hospital admissions due to complications related to congestive heart failure (CHF) among patients with compensated CHF [59]. Additionally, studies have consistently confirmed that both overt and subclinical hypothyroidism are correlated with a heightened risk of all-cause mortality, cardiac mortality, and hospital admissions in individuals with CHF [60]. According to our study, 61 out of 200 patients (30.5%) showed thyroid dysfunction, with 27 patients (13.5%) exhibiting hypothyroidism. Among the patients who passed away, 23 out of 73 (31.5%) had thyroid diseases. In contrast to the results obtained from the studies we presented concerning deceased patients with advanced heart failure, we did not observe any statistically significant distinction in the mortality rate among the subgroups (*p* = 0.81).

The prevalence of thyroid dysfunction in the adult population varies significantly, contingent upon the methodology employed for diagnosis, the criteria defining it, the individual iodine consumption, and whether the assessment of thyroid dysfunction is conducted within the parameters of overt thyroid diseases such as hypothyroidism or primary hyperthyroidism, or in the context of subclinical thyroid disease [61,62]. Within the Introduction Section of our article, we have provided the definitions that served as the basis for our classification of patients into distinct subgroups. Thyroid disorders display a higher prevalence among women compared to men, with hyperthyroidism’s prevalence estimated from 0.5% to 2%, with a tenfold higher occurrence in females than in males in areas with ample iodine [11,63]. The prevalence of subclinical hyperthyroidism ranges from 0.5% to 6.3%, showing a higher incidence in individuals aged 65 years and older [64,65]. In regions with adequate iodine levels, spontaneous hypothyroidism’s prevalence ranges from 1% to 2%, with a higher frequency in elderly females, occurring ten times more often in females than in males. The prevalence of subclinical hypothyroidism is higher, reaching 10% among individuals aged over 60 years. In our study, 46.9% (15/32) of female patients had thyroid dysfunction, whereas only 27.4% (46/168) of male patients exhibited the same condition. This gender-based difference in the incidence of thyroid dysfunction was statistically significant, with a *p*-value of 0.028. The findings of our study reveal that hyperthyroidism is significantly prevalent among women who suffer from advanced heart failure. In fact, a notable 40.6% (13 out of 32) of the study participants were diagnosed with this condition. The prevalence of hypothyroidism was significantly lower in the women’s group, with an incidence rate of merely 6.25%. In the male subpopulation, the prevalence of hyperthyroidism was higher than that of hypothyroidism, with 32 out of 46 cases compared to 14 out of 46, resulting in a *p*-value of 0.00014. In the clinical domain, the most common TH metabolism alteration is the low T3 syndrome, observed in 15% to 30% of HF patients, with incidence varying based on the clinical severity of the disease. In contrast, subclinical hypothyroidism occurs in 6% and 3% of HF patients, respectively [57].

It is recommended to assess the risk of amiodarone-induced thyroid dysfunction before initiating prolonged amiodarone therapy. Patients receiving amiodarone treatment should undergo thyroid function monitoring every six months, accompanied by regular electrocardiogram (ECG) evaluations. The disorders of thyroid function manifest in 10.3–14.7% of patients undergoing chronic amiodarone therapy for ventricular arrhythmias and atrial fibrillation (AF) [32,66]. In our population comprising patients with advanced heart failure, thyroid dysfunction was present in 30.5% of individuals receiving chronic amiodarone treatment. It is interesting to note that in the male subpopulation, hyperthyroidism induced by amiodarone was predominant, highlighting this subgroup of patients as the most vulnerable to thyrotoxicosis. Interestingly, we did not observe any female patients with AIH. However, this observation may be associated with the relatively small number of patients and the underrepresentation of women in the advanced heart failure population.

### Limitations

Due to various confounding factors, we did not perform an analysis of the hospitalization rate. The confounding factors included pre-planned admissions for check-ups, the management of implantable cardiac devices, the management of non-cardiac-related comorbidities, and the unexpected extension of hospital stays due to non-medical reasons. The dynamic nature of TSH, FT4, and FT4 levels in the course of hypo- or hyperthyroidism treatment posed challenges in reliable quantification of their influence on the survival of these heart failure patients. Furthermore, the precise timing of amiodarone initiation within the pharmacotherapy regimen could not always be determined accurately. Many patients were admitted to our center with medication established at various medical facilities, often with incomplete medical records.

## 5. Conclusions

Thyroid dysfunction is a relatively common abnormality among patients with heart failure; in about one-third of them, it is caused by antiarrhythmic treatment. For patients with heart failure, the dominant clinical manifestation of thyroid dysfunction is hyperthyroidism, whereby nearly 60% of the population with heart failure and hyperthyroidism have their conditions resulting from the usage of amiodarone. Further exploration into the management of thyroid dysfunction within the context of heart failure seems of important value.

## Figures and Tables

**Table 1 biomedicines-12-00567-t001:** Baseline clinical characteristics of the studied group.

Clinical Data	
Age, years [SD]	56.5 (±10)
Gender, female, N (%)	32 (16)
Long-term mechanical circulatory support, N (%)	30 (15)
Implantable cardioverter-defibrillator, N (%)	194 (97)
Ischemic etiology, N (%)	110 (55)
Dilated cardiomyopathy, N (%)	80 (40)
Valvular cause of heart failure, N (%)	6 (3)
Other	4 (2)
NYHA	
NYHA I, N (%)	1 (0.5)
NYHA I/II, N (%)	2 (1)
NYHA II, N (%)	104 (52)
NYHA II/III, N (%)	11 (5.5)
NYHA III, N (%)	74 (37)
NYHA III/IV, N (%)	2 (1)
NYHA IV, N (%)	6 (3)
Chronic kidney disease, grade 3, N (%)	73 (36.5)
Chronic kidney disease, grade ≥ 4, N (%)	5 (2.5)
Creatinine level, [μmol/L], mean [SD]	114 [36]
Diabetes, N (%)	51 (25.5)
Ventricular arrhythmia, N (%)	117 (58.5)
Sustained tachycardia, N (%)	94 (47)
Non-sustained VT, N (%)	88 (44)
Ventricular fibrillation (aborted), N (%)	16 (8)
Body mass index (kg/m^2^), [SD]	28.6 [5.7]
NTproBNP (pg/mL), mean [SD]	3947 [4807]
TSH (μIU/mL), mean [SD], patients without thyroid dysfunction, 139 pts	1.175 [1.12]
FT3 (pg/mL), mean [SD], patients without thyroid dysfunction, 139 pts	2.49 [1.11]
FT4 (pg/mL), mean [SD], patients without thyroid dysfunction, 139 pts	1.12 [0.23]
Amiodarone treatment, patients without thyroid dysfunction, 139 pts, N (%)	35 (25.2)
Thiamazole treatment, patients without thyroid dysfunction, 139 pts, N (%)	0 (0)
Thyroid hormone supplementation, patients without thyroid dysfunction, 139 pts, N (%)	0 (0)
TSH (µIU/mL), mean [SD], patients with diagnosed hyperthyreosis, 34 pts	3.64 [5.1]
FT3 (pg/mL), mean [SD], patients with diagnosed hyperthyreosis, 34 pts	2.37 [0.55]
FT4 (pg/mL), mean [SD], patients with diagnosed hyperthyreosis, 34 pts	1.12 [0.19]
Amiodarone treatment, patients with diagnosed hyperthyreosis, 34 pts, N (%)	7 (20.6)
Thiamazole treatment, patients with diagnosed hyperthyreosis, 34 pts, N (%)	15 (44.1)
Thyroid hormone supplementation, patients with diagnosed hyperthyreosis, 34 pts, N (%)	0 (0)
TSH (µIU/mL), mean [SD], patients with diagnosed hypothyroidism, 27 pts	4.08 [3.4]
FT3 (pg/mL), mean [SD], patients with diagnosed hypothyroidism, 27 pts	2.12 [0.57]
FT4 (pg/mL), mean [SD], patients with diagnosed hypothyroidism, 27 pts	1.1 [0.18]
Amiodarone treatment, patients with diagnosed hypothyroidism, 27 pts, N (%)	7 (25.9)
Thiamazole treatment, patients with diagnosed hypothyroidism, 27 pts, N (%)	0 (0)
Thyroid hormone supplementation, patients with diagnosed hypothyroidism, 27 pts, N (%)	25 (92.6)

Abbreviations: NTproBNP—N-terminal prohormone of brain natriuretic peptide; NYHA—New York Heart Association; pts—patients; FT3—Free triiodothyronine; FT4—Free thyroxine; TSH—thyroid-stimulating hormone; and VT—ventricular tachycardia.

**Table 2 biomedicines-12-00567-t002:** Mortality related to thyroid dysfunction. No statistically significant differences were observed between subgroups.

Patients	Total	Survivors, N (%)	Deceased, N (%)	*p*
Hyperthyroidism	34	22 (64.7)	12 (35.3)	0.94
Hypothyroidism	27	16 (59.3)	11 (40.7)	0.64
No thyroid dysfunction	139	89 (64)	50 (36)	-

## Data Availability

The original contributions presented in the study are included in the article, further inquiries can be directed to the corresponding author.

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
