# Peer review of "Thyroid Dysfunction in Advanced Heart Failure Patients and Its Correlation with Amiodarone Therapy: A Two-Year Study"

_biomedicines, 2024, doi:10.3390/biomedicines12030567_

Round 1
Reviewer 1 Report
Comments and Suggestions for Authors
This is an important registry study on impact of thyroid dysfunction on prognosis in advanced heart failure patients. Thyroid disfunction was found in 61 from 200 patients, more in women. That was unexpected, the high rate of hyperthyroidism was observed, mainly subclinical. Thus only 15 patients required thyrosostatics.
Several corrections or clarification are required.
1. Clarification of the TSH study date is required. Namely patients with thyroid dysfunction at inclusion were analyzed or any alteration of TSH during follow-up.
2. Clinical characteristics, including amiodarone use, levothyroxine use and thyrosol use, as so as TSH level in patients with hypo, hyper and normal thyroid function should be presented. In table 1 for ex.
3. If basal TSH were utilized as group determinator, survival analysis is required to support the influence (or lack of it) of thyroid function on prognosis. Moreover, significance of clinical thyroid disease that required levothyroxine or thyrosol, could be analyzed separately
4. The association between thyroid dysfunction and morbidity (hospitalization rate) could be analyzed or mentioned in limitation section.
Comments on the Quality of English LanguageEditing is required
Author Response
Reviewer #1
Thank you for your review and valuable comments. We corrected the mistakes and inaccuracies and modified the text according to your suggestions.
The comments made by the reviewers have been addressed in this document. The modifications have been highlighted in yellow font in the manuscript's main text.
Please find below the point by point responses.
This is an important registry study on impact of thyroid dysfunction on prognosis in advanced heart failure patients. Thyroid disfunction was found in 61 from 200 patients, more in women. That was unexpected, the high rate of hyperthyroidism was observed, mainly subclinical. Thus only 15 patients required thyrosostatics.
Several corrections or clarification are required.
- Clarification of the TSH study date is required. Namely patients with thyroid dysfunction at inclusion were analyzed or any alteration of TSH during follow-up.
Ad. 1 In the methodology section, we specified that hormone levels were analyzed from admission.
- Clinical characteristics, including amiodarone use, levothyroxine use and thyrosol use, as so as TSH level in patients with hypo, hyper and normal thyroid function should be presented. In table 1 for ex.
Ad. 2 We developed it in Table No 1 in order to make it more transparent.
- If basal TSH were utilized as group determinator, survival analysis is required to support the influence (or lack of it) of thyroid function on prognosis. Moreover, significance of clinical thyroid disease that required levothyroxine or thyrosol, could be analyzed separately
Ad. 3 The prior clinical diagnosis served as the group determinant, with laboratory diagnostics conducted for each patient as additional factors. Some patients required Euthyrox or Thyrosol intermittently, as indicated by their medical history. We considered the data collected at the time of enrollment.
- The association between thyroid dysfunction and morbidity (hospitalization rate) could be analyzed or mentioned in limitation section.
Ad. 4 We addressed this issue in the limitations section, noting that patients with advanced heart failure are hospitalized not solely for HF decompensation. Certain procedures, typically conducted on an outpatient basis, are also performed during hospital stays in this patient cohort (e.g., colonoscopy). Additionally, procedures such as ICD implantation/exchange are performed during hospitalization.
Once again thank you for your time and contribution devoted to assessing our manuscript.
Yours Sincerely,
Authors

Reviewer 2 Report
Comments and Suggestions for Authors
Dear Authors,
a few thoughts about the study in question.
The study “Thyroid Dysfunction in Severe Heart Failure Patients and its Correlation with Amiodarone Therapy: A Two-Year Study” deals with a topic of interest for which some literature data exist, although in less severe populations.
I take my cue from the title to specify that the term "severe" cannot be used as a synonym for
‘Advanced HF’ possibly also defined as 'refractory', or 'end-stage' heart failure.
“We enrolled 200 patients diagnosed with advanced heart failure according to the current ESC criteria (35,36). The patients were admitted consecutively between December 31, 2018 and December 31, 2019”: in this regard, bibliography entry #35 refers to 2021 guidelines therefore subsequent to the enrollment period.
“The data were acquired from patients ’clinical documentation during hospital stays, ambulatory visits and cardioverter-defibrillator controls”. Considering that the data were also collected via ICDs, it would be necessary to know how many patients were ICD carriers.
The definition of hyperthyroidism and hyperthyroidism has been correctly reported, while the term "thyrotoxicosis" is also reported in the paper. It would be necessary to know if the terms hyperthyroidism and thyrotoxicosis have been used synonymously; in the latter case, even if formally incorrect, it should still be specified.
Since “30 patients underwent heart transplantation (15%)” should be specified whether the transplanted subjects were included or excluded from the analysis or considered only for part of the analysis.
In addition, Table 1 shows that they have been subjected to “Long-term mechanical circulatory support, N 30 (15%)” patients. This number is exactly the same as the number of transplanted patients in the text. It is necessary to clarify this point.
Also, in Table 1, the percentage sum of patients per NYHA class is far from 100%.
“Treatment protocols for all patients adhered to the current guidelines set by the European Society of Cardiology for the treatment of heart failure 164 (1,2)”, the bibliography is incorrect as #1, published in 2021, does not refer to guidelines for the treatment of HF while #2 deals with a completely different topic.
The following section is not clear: “Thyroid hormone supplementation was required in 12.5% patients and 7.5% required application of thiamazole…8.5% of patients had TSH outside of the 0.35-4.94 177 μIU/ml range”: 20% of patients treated for thyroid dysfunction but only 8.5% with out-of-range TSH?
“In total, 61 (30.5%) patients exhibited thyroid dysfunction”: from the percentages shown in the lines preceding the statement, it is not clear how this number is reached.
“Thyroid dysfunction was present among…27.4% (46/168) men; … In the male subpopulation, hyperthyroidism was more prevalent at 23/46 compared to hypothyroidism at 14/46; P=0.00014.” The sum of subjects with hypothyroidism (14) and hyperthyroidism (23) is far from the total number of males with thyroid dysfunction (reported as 46).
“In total, 34 patients were diagnosed with hyperthyroidism, with 14 having hyperthyroidism not directly linked to the application of amiodarone and 20 as a consequence of administering amiodarone (30.1% of patients receiving chronic treatment with amiodarone).”: it is not clear to which population the 30.1% percentage refers.
The discussion section is long and largely referable to literature data and not to a real discussion of the study in question.
It lacks a section dedicated to the limitations of the study.
In view of the size of the sample examined, the conclusions cannot be categorical but only possibilist.
Comments on the Quality of English LanguageIt is not fluent English, in some sentences it must be reviewed.
Author Response
Thank you for your review and valuable comments. We corrected the mistakes and inaccuracies and modified the text according to your suggestions.
The comments made by the reviewers have been addressed in this document. The modifications have been highlighted in yellow font in the manuscript's main text.
Please find below the point by point responses.
Reviewer #2
- The study “Thyroid Dysfunction in Severe Heart Failure Patients and its Correlation with Amiodarone Therapy: A Two-Year Study” deals with a topic of interest for which some literature data exist, although in less severe populations.
I take my cue from the title to specify that the term "severe" cannot be used as a synonym for
‘Advanced HF’ possibly also defined as 'refractory', or 'end-stage' heart failure.
Ad. 1 As suggested by the Reviewer, we changed the terminology "severe" to advanced/refractory/end-stage HF as all of our patients were evaluated for advanced heart failure therapies such as cardiac transplantation and long-term mechanical circulatory support (MCS) ).
- “We enrolled 200 patients diagnosed with advanced heart failure according to the current ESC criteria (35,36). The patients were admitted consecutively between December 31, 2018 and December 31, 2019”: in this regard, bibliography entry #35 refers to 2021 guidelines therefore subsequent to the enrollment period.
Ad. 2 It was our intention to convey that patients also met the criteria for advanced heart failure according to the 2021 criteria; however, as suggested by the Reviewer, we only referred to the guidelines available at the time of data collection.
- “The data were acquired from patients ’clinical documentation during hospital stays, ambulatory visits and cardioverter-defibrillator controls”. Considering that the data were also collected via ICDs, it would be necessary to know how many patients were ICD carriers.
Ad. 3 The exact number of ICD carriers can be found in Table 1.
- The definition of hyperthyroidism and hyperthyroidism has been correctly reported, while the term "thyrotoxicosis" is also reported in the paper. It would be necessary to know if the terms hyperthyroidism and thyrotoxicosis have been used synonymously; in the latter case, even if formally incorrect, it should still be specified.
Ad. 4 We added the definitions of hyperthyroidism and thyrotoxicosis after De Leo et al. in the Introduction.
- Since “30 patients underwent heart transplantation (15%)” should be specified whether the transplanted subjects were included or excluded from the analysis or considered only for part of the analysis.
- In addition, Table 1 shows that they have been subjected to “Long-term mechanical circulatory support, N 30 (15%)” patients. This number is exactly the same as the number of transplanted patients in the text. It is necessary to clarify this point.
Ad. 5 & 6 We corrected the data in the Result section: 48 patients underwent heart transplantation (24%). We also clarified that Patients who underwent heart transplantation were analyzed in the study up to the time of transplantation.
- Also, in Table 1, the percentage sum of patients per NYHA class is far from 100%.
- “Treatment protocols for all patients adhered to the current guidelines set by the European Society of Cardiology for the treatment of heart failure 164 (1,2)”, the bibliography is incorrect as #1, published in 2021, does not refer to guidelines for the treatment of HF while #2 deals with a completely different topic.
Ad. 7&8 There were errors in the text editing process. We corrected it.
- The following section is not clear: “Thyroid hormone supplementation was required in 12.5% patients and 7.5% required application of thiamazole…8.5% of patients had TSH outside of the 0.35-4.94 177 μIU/ml range”: 20% of patients treated for thyroid dysfunction but only 8.5% with out-of-range TSH?
Ad. 9 Some patients were admitted with a pre-existing diagnosis and treatment regimen initiated prior to hospitalization. These patients received thiamazole or thyroid hormone supplementation, and likely due to previous therapeutic intervention, their levels of TSH, FT3, and FT4 were within normal ranges.
- “In total, 61 (30.5%) patients exhibited thyroid dysfunction”: from the percentages shown in the lines preceding the statement, it is not clear how this number is reached.
Ad.10 We went into detail with these data in Table 1 to make it more transparent.
11.“Thyroid dysfunction was present among…27.4% (46/168) men; … In the male subpopulation, hyperthyroidism was more prevalent at 23/46 compared to hypothyroidism at 14/46; P=0.00014.”The sum of subjects with hypothyroidism (14) and hyperthyroidism (23) is far from the total number of males with thyroid dysfunction (reported as 46).
Ad.11 There was a mistake in copying. Instead of 23, it should be 32. It is now corrected.
- “In total, 34 patients were diagnosed with hyperthyroidism, with 14 having hyperthyroidism not directly linked to the application of amiodarone and 20 as a consequence of administering amiodarone (30.1% of patients receiving chronic treatment with amiodarone).”: it is not clear to which population the 30.1% percentage refers.
Ad. 12 The study found that out of a total of 65 heart failure patients who received amiodarone, 34 were diagnosed with hyperthyroidism. Of these 34 patients, 14 were diagnosed with hyperthyroidism that was not directly linked to the use of amiodarone, while the remaining 20 patients were diagnosed with hyperthyroidism as a direct consequence of administering amiodarone. It is noteworthy that the latter group constitutes 30.1% of the total population of heart failure patients who received amiodarone.
- The discussion section is long and largely referable to literature data and not to a real discussion of the study in question. It lacks a section dedicated to the limitations of the study.
In view of the size of the sample examined, the conclusions cannot be categorical but only possibilist.
Ad. 13 We discussed the limitations of our study and highlighted the need for further research due to the small size of the study group.
Once again thank you for your time and contribution devoted to assessing our manuscript.
Yours Sincerely,
Authors
Round 2
Reviewer 1 Report
Comments and Suggestions for Authors
The manuscript is improved and can be recommended to the Journal
Comments on the Quality of English LanguageEnglish is fine
Reviewer 2 Report
Comments and Suggestions for Authors
Dear Authors,
the study is interesting and the constructive proposals for revision have been fully accepted.
Comments on the Quality of English LanguageMinor editing.